# Comprehensive Phenotyping in Inflammatory Bowel Disease: Search for Biomarker Algorithms in the Transkingdom Interactions Context

**DOI:** 10.3390/microorganisms10112190

**Published:** 2022-11-04

**Authors:** Ayelén D. Rosso, Pablo Aguilera, Sofía Quesada, Florencia Mascardi, Sebastian N. Mascuka, María C. Cimolai, Jimena Cerezo, Renata Spiazzi, Carolina Conlon, Claudia Milano, Gregorio M. Iraola, Alberto Penas-Steinhardt, Fiorella S. Belforte

**Affiliations:** 1Laboratorio de Genómica Computacional (GeC-UNLu), Departamento de Ciencias Básicas, Universidad Nacional de Luján, Luján 6700, Argentina; 2Programa del Estudio de Comunicación y Señalización Interreino (PECSI-UNLu), Departamento de Ciencias Básicas, Universidad Nacional de Luján, Luján 6700, Argentina; 3Consejo Nacional de Investigaciones Científicas y Técnicas (CONICET), Ciudad Autónoma de Buenos Aires C1425FQB, Argentina; 4Instituto de Ecología y Desarrollo Sustentable (INEDES-CONICET-UNLu), Departamento de Ciencias Básicas, Universidad Nacional de Luján, Luján 6700, Argentina; 5Instituto de Medicina Traslacional e Ingeniería Biomédica (IMTIB), CONICET, Instituto Universitario del Hospital Italiano (IUHI), Hospital Italiano de Buenos Aires (HIBA), Ciudad Autónoma de Buenos Aires C1199, Argentina; 6Servicio de Gastroenterología, Hospital Nacional Prof. Alejandro Posadas, Ciudad Autónoma de Buenos Aires 1704, Argentina; 7Laboratorio de Genómica Microbiana, Institut Pasteur Montevideo, Montevideo 11400, Uruguay; 8Centro de Biología Integrativa, Universidad Mayor, Santiago 7510041, Chile; 9Wellcome Sanger Institute, Wellcome Genome Campus, Cambridgeshire CB10 1SA, UK; 10Instituto Universitario de Ciencias de la Salud, Fundación H.A. Barceló, Ciudad Autónoma de Buenos Aires 1127, Argentina

**Keywords:** comprehensive-phenotyping, gut-microbiota, ulcerative-colitis, crohn-disease

## Abstract

Inflammatory bowel disease (IBD) is the most common form of intestinal inflammation associated with a dysregulated immune system response to the commensal microbiota in a genetically susceptible host. IBD includes ulcerative colitis (UC) and Crohn’s disease (CD), both of which are remarkably heterogeneous in their clinical presentation and response to treatment. This translates into a notable diagnostic challenge, especially in underdeveloped countries where IBD is on the rise and access to diagnosis or treatment is not always accessible for chronic diseases. The present work characterized, for the first time in our region, epigenetic biomarkers and gut microbial profiles associated with UC and CD patients in the Buenos Aires Metropolitan area and revealed differences between non-IBD controls and IBD patients. General metabolic functions associated with the gut microbiota, as well as core microorganisms within groups, were also analyzed. Additionally, the gut microbiota analysis was integrated with relevant clinical, biochemical and epigenetic markers considered in the follow-up of patients with IBD, with the aim of generating more powerful diagnostic tools to discriminate phenotypes. Overall, our study provides new insights into data analysis algorithms to promote comprehensive phenotyping tools using quantitative and qualitative analysis in a transkingdom interactions network context.

## 1. Introduction

Inflammatory bowel disease (IBD) represents a complex, polygenic chronic disorder of unknown etiology [1]. It is estimated that IBD is associated with industrialized countries, where the decrease in contact with microorganisms, parasites or their derivatives promote an increase in the prevalence of chronic inflammatory diseases. This is consistent with the hygiene hypothesis that suggests that a lack of early childhood exposure to pristine microbial conditions may increase the individual’s susceptibility to disease [2]. In particular, IBD includes two main phenotypes: Crohn’s disease (CD) and ulcerative colitis (UC). However, between 10–17% of IBD patients do not have a definitive diagnosis of CD or UC, a phenomenon known as “inflammatory bowel disease unclassified” (IBDU) [3]. UC mainly comprises the rectum, affecting contiguously and symmetrically the colon being more severe distally. Depending on the colonic segments involved, UC extent can be classified as proctitis, left-sided colitis, or extensive colitis. CD is not continuous or symmetrical and usually does not involve the rectum. CD is also associated with intestinal granulomas, strictures and fistulas, which are not common findings in UC [1]. In CD, inflammation is often transmural, whereas in UC is typically confined to the mucosa. As eventually CD could compromise the colon and share UC pathognomonic manifestations, its diagnosis could be confused despite being different diseases. Colonic CD and UC need to be distinguished by differences in genetic predisposition markers, risk factors and clinical features [1,4]. Additionally, IBD can be identified months or years after the first appearance of symptoms, requiring clinical, serological, radiological, endoscopic and histological information to define its prognosis and treatment.

Globally, IBD is the most common form of intestinal inflammation associated with a deregulated immune-system response to commensal microbiota in a genetically susceptible host. Multicellular organisms, such as humans, rely heavily on their commensal, symbiotic microbiota. This heterogeneous community is composed of microbial groups such as viruses, bacteria, archaea, fungi, and other eukaryotes found in multiple body niches, such as the intestine, skin, vagina, mouth, etc. In particular, the human gut microbiota outnumbers human cells and expresses more genes than those present in our genome [5]. These complex communities of microorganisms mediate physiologically important chemical transformations playing a key role in recovering energy and nutrients from the diet as well as promoting ion absorption at the colon level [5].

Likewise, it is postulated that there is an intestinal “inter-kingdom” communication mediated, at least in part, by microvesicles carrying different types of biological messages. Among the known nanometric-sized microvesicles are exosomes, which play a fundamental role in RNA-mediated cell-to-cell communication, especially in inflammatory and malignant processes. Currently, there are many studies that propose the search for miRNAs as plasmatic and fecal biomarkers associated with chronic inflammatory processes [6,7]; however, few describe its interaction with the microbiome and possible inter-kingdom communication mechanisms [8]. In this sense, it is essential to assess the composition of the gut-associated microbiota in the context of IBD, as certain groups of organisms may alter the communication between the immune system and commensal microbes, triggering an exacerbated response in the intestinal mucosa [9].

However, little is known about the human microbiome of South American populations [10,11,12]. Reports of gut dysbiosis processes associated with IBD in this region are almost null, despite reports of a rapid increase in the incidence in South America [2]. So far, most of the available literature on the study of the gut microbiome associated with IBD points to developed countries, which differ both in the genetic background and in several environmental factors from our population [13]. In particular, our group published the microbial gut diversity of Buenos Aires (BA), being the first local report in this area [14]. Since BA is the second most populated agglomeration in South America and the southern hemisphere (with a large genetic and cultural component of European immigration interacting with local indigenous people), BA microbiota analysis was compared with different 16S rRNA gene sequence data sets. In addition, our group has recently performed a two-time point analysis of the fecal microbiota in those Metropolitan Buenos Aires inhabitants previously studied to compare pre-pandemic data and its variation during preventive and compulsory social isolation (PCSI) in 2020 [15]. These works provided the first data related to the gut microbiota of our population and its resilience to disturbances.

In the present work, we characterized, for the first time in our region, epigenetic biomarkers and gut microbial profiles associated with UC and CD patients in the BA Metropolitan area and revealed differences between non-IBD controls and IBD patients. General metabolic functions associated with the gut microbiota, as well as core microorganisms within groups, were also analyzed. Additionally, the gut microbiota analysis was integrated with relevant clinical, biochemical, and epigenetic markers considered in the follow-up of patients with IBD, with the aim of generating more powerful diagnostic tools to discriminate phenotypes. [16]. In this sense, there is an enormous unsatisfied need for biological markers that allow evidence of the status, progression and response to therapy of complex diseases with an inflammatory component. Defining data analysis algorithms that integrate clinical, biochemical, and metagenomic information is relevant to facilitate patient evaluation and allow the discovery of new biomarkers. Overall we intend to generate comprehensive phenotyping tools providing new knowledge on the differences and similarities of the gut microbiota of IBD patients compared to controls of the South American population, using quantitative and qualitative analysis of microbiome profiles as well as clinical, biochemical and epigenetic parameters, in a context of transkingdom interaction.

## 2. Materials and Methods

### 2.1. Ethics Statement

This study received approval from Hospital Nacional Profesor Alejandro Posadas according to local regulations and the Helsinki declaration. Written informed consent was obtained from all study participants.

### 2.2. Selection of Participants and Environmental Data

This cross-sectional study recruited consecutive UC and CD patients attending the Gastroenterology Service of the Posadas Hospital. Non-IBD controls were selected according to age and body mass index (BMI) in order to match the patient population characteristics (Table 1), considering the same geographical location for all participants. The exclusion criteria established for both non-IBD controls and IBD patients considered individuals who have not received antibiotic therapy in the last 6 months, subjects on extreme diets (e.g., macrobiotics, vegans), surgical intervention in the gastrointestinal tract (gastrectomy, bariatric surgery), pregnancy, digestive neoplasias, patients on renal replacement therapy, transplant recipients and HIV infected. Additional exclusion criteria for non-IBD controls were the presence of IBD or irritable bowel syndrome (IBS). The sample size was determined using a Dirichlet multinomial distribution model. With an expected number of more than 50,000 sequence reads per sample and an α of 5%, ~15 subjects per group were required for ~80% power [17].

### 2.3. General Diagnosis

Physicians apply the European Crohn’s and Colitis Organization (ECCO) guidelines for the diagnosis, treatment and surveillance of IBD patients. The diagnosis was performed by combined studies of endoscopy, histology, serology, and clinical data. UC patients were defined based on Truelove & Witts’ criteria for clinical activity [18]. On such a basis, patients can be classified as normal, mild, moderate or severe. Endoscopic studies were evaluated based on Mayo Score, as follows: normal, mild disease (erythema, decreased vascular pattern, mild friability), moderate disease (marked erythema, lack of vascular pattern, friability, erosions), or severe disease (spontaneous bleeding, ulceration) [19,20]. CD patients were characterized based on Crohn’s Disease Activity Index (CDAI) for clinical activity [21]. For endoscopic evaluations, the Simple Endoscopic Score for Crohn’s Disease (SES-CD) was defined [22]. The general diagnosis for active or remission patients was defined by clinical and endoscopic scores together; this determination is based on medical criteria. To assess the general information about participants, anthropometric measurements (height, weight, and waist circumference) and blood pressure were determined by standardized protocols. Body mass index (BMI) was calculated as weight (kg)/[height(m)]^2^.

### 2.4. Biochemical Measurements

After a 12 h overnight fast, venous blood samples were obtained from volunteers for further analyses. Fasting plasma glucose (FPG), creatinine, total cholesterol, triglycerides, LDL-C, and HDL-C were determined by enzymatic methods in serum samples using standardized procedures. Hemoglobin, platelets, albumin, and ultra-sensitive C-Reactive-Protein (us-CRP) were measured in order to determine the severity of the disease. All biochemical measurements were performed at the Hospital Posadas Laboratory, BA, Argentina. Given the low predictive nature of serological data in IBD patients, anti-Saccharomyces cerevisiae/antineutrophil cytoplasmic antibodies (ASCA/ANCA antibodies) were determined by immunofluorescence and immunoenzymatic standard techniques only in those patients who required it to define the diagnosis. Likewise, patients undergo annual video colonoscopies to monitor the pathology, as well as histopathological analysis of biopsy samples taken for this purpose.

### 2.5. Blood and Stool RNA Extraction

In order to investigate the expression levels of hsa-miR-146a-5p, hsa-miR-155-5p, and hsa-miR-223-3p IBD biomarkers, 2 types of samples were chosen: (BCF) and fecal cell fraction (FCF). Briefly, the BCF isolation was performed at the time of sampling by density gradient by means of centrifugation at 1600 rpm (20 min) of the blood diluted in half with physiological solution (SF: NaCl 0.9% M/V) on Ficoll Hypaque reagent. BCF was resuspended in 1 mL of Trizol reagent and stored at −80 °C for further processing. To generate FCF, 250 mg of solid feces were transferred in 2 mL tubes. Filtered 1× PBS was added to the tubes until completing the 2 mL of the tube and vortexed for 2 min. Tubes were centrifuged at 100 g for 15 min in order to eliminate macroscopic traces of food from the sample, recovering the supernatant in a new tube. The supernatants were transferred to 2 mL tubes to be centrifuged at 8000 g for 5 min. The pellet FCF was resuspended in 1ml of Trizol reagent and stored at −80°C for further processing. Total RNA was extracted using the phase separation technique with Trizol reagent following the manufacturer’s instructions.

### 2.6. miRNAs Identification

Reverse transcription of hsa-miR-146a-5p, hsa-miR-155-5p, and hsa-miR-223-3p IBD biomarkers was performed using Stem Loop Primers (SLP) strategy and the reverse transcriptase enzyme MMLV (Moloney Murine Leukemia Virus Reverse Transcriptase) from Promega following the manufacturer’s instructions. SLPs were designed using the sequence described by Martha F. Kramer [23] and adding a 3’ end of six nucleotides sequence specific to each miRNA. The quantitative real-time PCR method (qPCR) was performed with “SYBR™ Select Master Mix” from Applied Biosystems in StepOnePlus Real-Time PCR equipment. Absolute Quantification by Standard Curve was implemented, using synthetic plasmids designed with the sequence corresponding to the PCR products that are expected to be obtained from the cDNAs generated by the SLPs strategy.

### 2.7. Stool Samples Collection and DNA Extraction

Each participant was given a written protocol for stool sample collection, which considered the introduction of approximately 5g of stool into a sterile wide-mouth tube containing a teaspoon in the lid. DMSO/EDTA/saturated sodium chloride buffer (DESS) was used to preserve the samples at both room and subzero temperatures, preventing freezing and cell damage as described [14]. The samples were aliquoted and stored at −80 °C until use. DNA extraction was performed using QIAamp PowerFecal DNA Kit following the manufacturer’s instructions. The concentration and purity of the nucleic acids were determined by spectrophotometry in NanoDrop ND-1000 (NanoDrop Technologies, Wilmington, DE, USA).

### 2.8. 16S Bacterial rRNA Fragment NGS

To amplify the 16S rRNA gene fragments of gut microbiota, 30 ng of purified DNA was used, and V3-V4 hypervariable regions of the bacterial 16S gene were amplified using 337F/805R primers. All segments of the variable regions of the 16S rRNA gene were normalized and multiplexed in a single tube. Sequencing was carried out using a MiSeq sequencer performing the synthesis sequencing methodology. Libraries were sequenced in the 5’ and 3’ directions (paired-end mode), ensuring 300–500 bp long sequences and ~150,000 average coverage for taxonomic identification.

### 2.9. Sequence Analysis and Comparison of Microbial Communities

Sequences generated were analyzed using Quantitative Insights Into Microbial Ecology (QIIME2) version 2021.2 software package [24]. Raw fastq reads were quality filtered (denoised, merged, and assessed for chimeras) to produce amplicon sequence variants (ASV) using the “DADA2” (Plugin version 2021.2.0) pipeline [25]. Figaro software was used to determine optimal trimming parameters (trunc-len for IBD patients and non-IBD control samples was f280 r200 around) [26]. After rare amplicon sequence variant filtering, a 0.1% minimum abundance filter was chosen based on the known 0.1% bleed-through between Illumina MiSeq runs [27,28], and tables were merged. In order to place each sequence into a phylogenetic reference tree, qiime fragment-insertion SEPP (version 4.3.10) was used (sepp-refs-silva-128.qza reference database) [29]. To perform the taxonomic classification by qiime feature-classifier classify-sklearn, we train a supervised learning classifier with RESCRIPt package [30,31], using the V3-V4 primers from this study and a 99% similarity threshold following the author’s tutorial. The database used for this taxonomic assignment was Silva Release 138 [32].

Alpha and beta diversity were calculated using qiime diversity core-metrics-phylogenetic pipeline at genus-level data. Alpha diversity measures were tested with a general linear model: Shannon diversity, which defines the total number of species (species richness) weighted for their relative abundances (species evenness). Differences in beta diversity were assessed using ADONIS permutation-based statistical test in vegan-R (accessed on 29 September 2022 from https://CRAN.R-project.org/package=vegan) implemented in QIIME2 (q2-diversity plugin 2021.2.0) [33]. A standard pipeline of Phylogenetic Investigation of Communities by Reconstruction of Unobserved States (PICRUSt version 2.4.1), implemented in QIIME2, was used to generate MetaCyc pathway ontology profiles [34,35]. Differences in taxa abundance at the genera level and functional profiles between groups were determined using the analysis of the composition of microbiomes (ANCOM) framework [36]. Core microbiota was defined as the set of amplicon sequence variants detected in 50–100% of the samples with a relative abundance threshold value above 0.01% (calculated with Core microbiome from R microbiome package). Data are presented using ggplot2 (version 3.3.1) with data extracted from QIIME2 artifacts by using qiime 2R (v0.99.5) (accessed on 29 September 2022 from https://github.com/jbisanz/qiime2R/). Posterior analysis was realized with phyloseq [version 1.34.0, [37]] and microbiome [version 1.12.0, [38]] R packages.

### 2.10. Logistic Regression Model

In order to achieve a predictive signature capable of discriminating between UC, CD, and non-IBD controls, forward stepwise logistic regression models were constructed using R version 4.0.5. Clinical characteristics age, BMI, biochemical parameters, core microbiota (identified by the microbiome R package), differentially abundant bacterial taxa (identified by the ANCOM framework) and statistically significant differentially abundant metabolic pathways (*p* < 0.05) (inferred by the PICRUSt pipeline) were used to design the best feature combination that could establish a predictive model for the disease. The global performance of each model was evaluated using the Area Under the ROC (AUROC).

### 2.11. Weighted Correlation Network Analysis

Considering that IBD affects and is caused by a wide range of molecular interactions, a holistic, comprehensive approach is required to identify key biomarkers involved. Clinical characteristics include age, BMI, biochemical measurements, miRNAs from BCF and FCF, core microbiome (identified by the microbiome R package), differentially abundant bacterial taxa (identified by the ANCOM framework) and statistically significant differentially abundant metabolic pathways (*p* < 0.05) (inferred by the PICRUSt pipeline) were input for weighted correlation network analysis in accordance with samples’ groups. To conduct this analysis, adjacency matrices based on Spearman’s correlation were built using the rcorr function in Hmisc R package version 4.7-0 [23,39]. Significant positive and negative correlations (*p* < 0.05) were selected to generate weighted correlation networks for CD and UC patients and non-IBD controls. Igraph R package version 1.2.7 (Csardi G, Nepusz T. The igraph software package for complex network research. InterJournal. 2006; Complex Systems 18: 1695) and Cytoscape version 3.71 were used for network visualization and topological characterization [40]. A combination of the centrality scores degree (the number of edges attached to a node) and eigenvector (the transitive influence of a node on high-score nodes) was applied to infer the regulatory relevance of each node and to identify potential keystones or hubs, which are nodes with the strongest influence on the environment’s dynamics [41,42]. The fast greedy modularity optimization algorithm of igraph was used to identify clusters within networks [43].

### 2.12. Data Accession

Raw sequences of the 16S rRNA gene reported in this article were deposited in NCBI Short Read Archive (SRA) and are accessible under the accession number PRJNA646271.

## 3. Results

### 3.1. Characteristics of the Studied Population

This cross-sectional study recruited unrelated individuals, including consecutive UC and CD patients attending the Gastroenterology Service of the Posadas Hospital. Non-IBD controls were selected according to age and body mass index (BMI) in order to match the patient population characteristics (Table 1), considering the same geographical location for all participants. All IBD patients were under chronic medical treatment as they had previously been diagnosed (Table 2).

Blood samples were taken by the responsible professionals during recruitment in order to determine biochemical variables and autoantibodies. These complementary parameters are useful for monitoring IBD patients since they can usually have low hemoglobin due to intestinal bleeding, hyperplatelet, low albumin levels, and high us-CRP, indicating inflammation and tissue damage. As shown in Table 2, we found no abnormal values among IBD patients in these parameters, except for the CD group whose us-CRP levels were high (>0.5 mg/dL).

In relation to the collected clinical data, most UC patients suffered from extensive or left-sided colitis, while one proctitis case. Notably, our UC population had most of the colon compromised, observing colonoscopic alterations from the rectum to the descending colon. In order to define our patients’ states, the clinical, endoscopic, and histological activity of our population were evaluated. These three parameters were considered since the disease may not manifest itself clinically but may be reflected in the endoscopic and histological examinations. In this sense, 25% of UC patients presented clinical activity classified with Truelove & Witts as mild-moderate phenotypes. Interestingly, 50% of the UC group showed the presence of inflammatory infiltrate in biopsy samples (Table 2). This also happens in patients with CD since most of them are under clinical remission, but almost 80% of the population shows inflammatory infiltration in histological studies.

### 3.2. miRNAs Characterization

The use of miRNAs as biomarkers is a widely studied strategy worldwide in various complex pathologies with an inflammatory component due to the availability of massive public data on gene expression [44,45,46,47,48]. In addition, as they regulate specific biological processes, their correlation with pathological phenotypes can be used to understand the molecular pathophysiology of diseases of unknown etiology, such as IBD. In order to analyze whether characteristic epigenetic biomarkers in IBD could provide valuable information in comprehensive phenotyping, miRNAs previously associated with IBD were selected [47] and quantified by qPCR both at the intestinal and peripheral levels. Figure 1 summarizes some of the significant differences observed in patients with UC and CD in the miRNAs studied: hsa-miR-146a-5p, hsa-miR-155-5p, and hsa-miR-223-3p. In BCF, a greater expression was quantified with statistical significance for hs-miR-155-5p in both UC and CD (*p* < 0.5), while in FCF, a significant increase in the expression of hs-miR-223-3p was observed in both groups (*p* < 0.05) (Figure 1A,D). In UC BCF, hs-miR-155-5p shows a significant increase in those treated with 5-AZA/steroids compared to controls (*p* < 0.05) (Figure 1B), being also increased in BCF of CD patients treated with 5-AZA (*p* < 0.05) (Figure 1C). It was also observed a significant increase of miR-155-5p in the FCF of patients with UC treated with 5-AZA (*p* < 0.05), while hs-miR-223-3p was significantly increased in the FCF of CD with the same treatment (*p* < 0.05; Figure 1E,F). There were no significant differences in the expression levels of hs-miR-146a-5p in patients with IBD in any of the samples evaluated (data not shown).

### 3.3. Microbial Composition

Low-quality reads and chimera sequences were filtered and denoised from the raw data with DADA2, eventually producing an average of 41.590 reads per sample. These reads corresponded to 1804 features and identified 222 genera of the Silva reference database. Approximately 70% of genera in non-IBD controls and IBD patients have a relative frequency between 0–5%. In contrast, the remaining 30% is represented by the genera *Bacteroides* in CD and *Prevotella* and *Bacteroides* in non-IBD controls and UC patients. All these genera belong to the phylum *Bacteroidota* (data not shown).

### 3.4. Alpha and Beta Diversity

The rarefaction curve came to a plateau indicating the sequencing depth was sufficient to measure the bacterial community (data not shown). Shannon diversity index was calculated [49], and its distribution was compared to determine if there were differences in the richness or uniformity of the samples between both groups. No significant differences were found between non-IBD controls and UC patients. In contrast, Shannon diversity values have been statistically different between non-IBD controls and CD (*p*-value < 0.05; Figure 2A,D). Hence Shannon diversity, chao1 richness, evenness, and phylogenetic diversity were similar between non-UC controls and UC patients. In terms of beta-diversity, robust compositional unweighted-weighted UniFrac distances were calculated and plotted with ellipses representing the 95% of confidence intervals of each group. Variability was explained by the first two principal components, and differences between non-IBD controls and UC patients were significant for unweighted diversity analysis (*p*-value < 0.05; Figure 2C), while the weighted diversity did not show significance (Figure 2B). On the other hand, when analyzing weighted and unweighted UniFrac distances between non-IBD controls and CD, both result in significance values (*p*-value < 0.05; Figure 2E,F).

### 3.5. Differentially Abundant Taxa between Patients and Controls

To study differentially abundant taxa between non-IBD controls and IBD patients, the compositional algorithm ANCOM-BC was used, with a 0.70 detection rate. Ten taxa were identified by the ANCOM framework: *Bifidobacterium* is the genera found differential within UC patients compared to non-IBD controls (Figure 3A). While *Bifidobacterium, Bacteroides, Lactobacillus*, *Streptococcus*, *Lachnoclostridium*, *Olsenella*, *Faecalibacterium*, *Clostridium_sensu_stricto_1*, *Flavonifractor*, *Turicibacter* are found to be differential only in CD compared to non-IBD controls with a *p*-value < 0.05 (Figure 3B).

### 3.6. Functional Analysis

To predict the functional capacity of the gut microbiota in non-IBD controls and IBD patients, a full pipeline of PICRUSt2 implemented in QIIME2 was used. PICRUSt uses an extended ancestral-state reconstruction algorithm to estimate which gene families are present and then combines gene families to estimate the composite metagenome. From the data on functional capabilities, we focused primarily on those which were associated with microbial metabolism. Significant differences were noticed in certain metabolic functions in the gut microbiota of non-IBD controls and IBD patients. MetaCyc Metabolic Pathways were predicted for groups, from which one only differs significantly in abundance for non-IBD controls, between UC patients and non-IBD controls (Figure 3C). For CD patients and non-IBD controls, thirteen metabolic pathways were significant, appearing in the group of CD patients (Figure 3D). Broadly, among gene families associated with metabolic pathways, menaquinol-9 biosynthesis (associated with the production of vitamin K2) was under-represented in UC patients. Instead, for CD patients, gene families associated with aromatics degradation appear in half of the differential metabolic pathways in this group when compared to non-IBD controls. Other annotations were not part of any specific route.

### 3.7. Common Core Microbiota

For non-IBD controls, 79 bacterial genera were identified as core microbiota, corresponding to 46.19% of the genera present in the group, 65 genera for UC representing 37.79% of the total, and 46 genera for CD core which correspond to 24.86% respectively (Figure 4A,B). The genera were assigned as core microbiota considering prevalence ≥50% and a detection threshold ≥0.10% (or else a frequency of ≥0.001). At the intersection of groups, 38 genera were found at the triple intersection, observing only a few core features exclusively represented in each of the groups. In this sense, there were 16 genera present in non-IBD controls that were lost in IBD patients’ core microbiota: three genera in UC and seven in CD specific to each group (Figure 4B). Clinical activity was also analyzed with respect to the core microbiota in IBD patients. The Venn diagram for UC activity is composed of the group of non-IBD controls and UC patients, classified as active or in remission, showing the number of genera common to and exclusive among these groups (Figure 4C). For the CD patients, the same approach was followed (Figure 4D).

### 3.8. Comprehensive Phenotyping Algorithms

In order to define data analysis algorithms that integrate clinical, biochemical, and metagenomic information to further gauge their ability to distinguish among patients with UC, CD, and non-IBD controls, several models were established applying forward stepwise logistic regression analyses and their potential predictive utility was assessed by ROC curve analysis.

A Biochemical Model (BM) was first developed using biochemical parameters, epigenetic biomarkers and clinical features, including age, BMI, BCF, and FCF miRNAs, FPG, creatinine, total cholesterol, triglycerides, LDL-C, HDL-C, hemoglobin, platelets, albumin, and *us-CRP*. As shown in Figure 5, the BM for CD was defined as follows (AUROC = 1): 208.69 + 0.001 × miR223_FCF − 0.015 × miR155_FCF − 1.187 × Platelets + 0.001 × miR155_BCF.

For the case of UC, the BM (Figure 6) was defined as follows (AUROC = 0.916): −6.077 + 3.024 × 10^−4^ × miR223_FCF + 1.721 × 10^−1^ × GOT + 2.894 × 10^−5^ × miR155_BCF.

Subsequently, we developed a Microbiota Model (MM) using the microbiota core data plus genera and metabolic pathways exhibiting statistically significant differences in the ANCOM analysis. A logistic regression model for CD (Figure 7) was defined as follows (AUROC =1): −51.66 + 1051745.67 × PWY5415 − 6208.25 × Ruminococcus_torques_group + 214212.44 × Hungatella.

On the other hand, MM for UC (Figure 8) was defined as follows (AUROC = 1): −89772 × Desulfovibrio − 43021 × Lachnoclostridium − 7759 × Subdoligranulum

### 3.9. Correlation Network Analysis

In the present work, we performed a correlation network analysis to gain insight into the interactions between the variables that are part of the selected logistic regression models for UC and CD patients. This aims to clarify which nodes hold the strongest influence on the others and to identify variables that could increase the specificity of the logistic regression models by means of taking part in the same cluster within the network.

The network representing the CD environment consisted of 129 nodes and 887 edges representing significant interactions (*p* < 0.05), out of which 23.11% were negative correlations. After ruling out those interactions that expressed Spearman’s correlation coefficients under |0.7|, 110 nodes and 252 edges were left (Figure 9). In order to identify potential keystones, we evaluated which nodes overpassed the obtained average for centrality scores, degree, and eigenvector (See Appendix A). The combination of these two situations was observed in the *Eubacterium siraeum group*, *Lachnospiraceae* GCA-900066575, and *Oscillospiraceae* UCG-002, suggesting them as keystones in CD patients.

The analysis of UC patients’ data resulted in a network consisting of 130 nodes and 638 edges representing significant positive correlations (*p* < 0.05), and following the filtering by Spearman’s coefficients, 90 nodes and 124 remained (Figure 10). The top three nodes exhibiting the highest degree and eigenvector centrality scores were *Ruminococcaceae* UCG-005, *Odoribacter*, and *Christensenellaceae* R7 group, identified as co-abundant as all three belonged to the same cluster after running the community detection algorithm.

The fast greedy algorithm, a hierarchical agglomeration algorithm for detecting community structure, was applied within the correlation networks consisting of nodes whose edges represented correlations greater than 0.7. This analysis resulted in 16 and 22 detected clusters in UC and CD networks, respectively (See Appendix A).

Figure 9 and Figure 10 show the clusters to which the variables from the respective logistic regression model belong, where it can be observed that most of these variables were distributed in different clusters, suggesting that each one could be representative of a distinctive biological process underlying the disease.

## 4. Discussion

The global epidemiology of inflammatory bowel disease (IBD) is changing rapidly with the increasing incidence and prevalence of the disease in developing regions. *Sriharan Selvaratnam* et al. performed a systematic review between January 1990 and December 2018 of the South American IBD burden and its rapid increase, particularly in industrialized regions [50]. They showed that there is a paucity of robust and representative epidemiological studies to explore modifiable risk factors and disease phenotypes in our region. Considering that the South American continent is home to more than 400 million people, it is expected to represent a significant proportion of those affected by IBD in the world. This translates into a notable diagnostic challenge, especially in underdeveloped countries where access to diagnosis or treatment is not always accessible for chronic diseases.

The present work characterized, for the first time in our region, epigenetic biomarkers and gut microbial profiles associated with UC and CD patients in the Buenos Aires Metropolitan area and revealed differences between non-IBD controls and IBD patients. General metabolic functions associated with the gut microbiota, as well as core microorganisms within groups, were also analyzed. Additionally, the gut microbiota analysis was integrated with relevant clinical, biochemical, and epigenetic markers considered in the follow-up of patients with IBD, with the aim of generating more powerful diagnostic tools to discriminate phenotypes.

In IBD, there are no good biomarkers with a strong positive predictive value. Complete blood count, erythrocyte sedimentation rate (ESR), us-CRP, albumin and total protein levels, fecal lactoferrin or calprotectin, and ANCA and ASCA antibodies may be helpful in diagnosis but are non-specific and non-pathognomonic [51]. In the present work, we sought to characterize three miRNAs in two different matrices in order to find biomarkers with greater predictive power and thus help in the diagnosis and prognosis of inflammatory bowel disease. The use of miRNAs as biomarkers is a widely studied strategy worldwide in various complex pathologies with an inflammatory component due to the availability of massive public data on gene expression [44,45,46,47,48]. Paraskevi et al. detected overexpression of hs-miR-155-5p in the peripheral blood of patients with active UC [52], while the expression in patients with active CD was invariable. In contrast, our patient population showed a significant increase in hs-miR-155-5p in the BCF of UC and CD patients. Pathak et al. identified SOCS1 (Cytokine Signaling Suppressor 1) as one of the targets of hs-miR-155 in intestinal myofibroblasts (IMF). Interestingly, some studies revealed that SOCS1 deficiency exacerbates intestinal inflammation and that SOCS1-deficient mice, which are hypersensitive to toll-like receptor (TLR) ligands, showing dysregulated cytokine production that perturbs immune cell activation and triggers the development of systemic autoimmunity suppressor of cytokine signaling-1 ameliorates dextran sulfate sodium-induced colitis in mice [53,54]. Furthermore, they found that the SOCS1 protein level is significantly lower in UC-derived IMF than in control-derived IMF. These findings suggest an inverse relationship between hs-miR-155 and SOCS1 expression. This points to SOCS1 as a powerful molecular switch that, by tuning key signaling pathways such as JAK kinases and activated cytokine receptor complexes, regulates the development of a variety of cell populations, inflammatory processes, and immune responses. Previous studies suggested that SOCS1 is inactivated in human IBD, contributing to dysregulated mucosal inflammation, although the cell populations involved were not identified [53].

When analyzing our patients according to their treatment, it was observed that although there is a general trend towards higher hs-miR155-5p expression in patients, the differences became significant in BCF of patients with UC treated with the 5-AZA-steroid combination and in the CD group treated with 5-AZA. This fact would show a limited effect of 5-AZA on the reduction of the inflammatory process, as it does not reduce the levels of this miRNA, which in this case acts as a buffer for the activity of the immune system. On the other hand, Daniel Szucs et al. found that hs-miR-155 expression was significantly increased in biopsies of macroscopically inflamed duodenal mucosa from pediatric patients compared with the control group [55,56]. Our results showed that this overexpression not only occurs at the peripheral level in UC BCF but also occurs in the fecal cell fraction FCF of adult patients, which broadens the potential of hs-miR-155-5p as a biomarker in our population.

Regarding hs-miR-223-3p, it was overexpressed in FCF of IBD patients in our population and is significantly increased in individuals with CD treated with 5-AZA and 5-AZA/steroids. This not only corroborates the ability of fecal hsa-miR 223-3p to discriminate IBD patients from controls but also highlights the limited role of 5-AZA in our CD population, which in many cases is the only treatment available due to the inability of low-income patients to access biologic treatments such as infliximab, adalimumab, golimumab, and vedolizumab. The hs-miR-223 is expressed and strictly regulated in hematopoietic cells contributing to a negative feedback mechanism that controls excessive innate immune responses in maintaining myeloid cell homeostasis [57]. There are many genes regulated by this miRNA, among them TRIL (toll-like receptors, TLR4) interactor with leucine-rich repeats), a key player in the costimulatory complex of TLR4 [58]. It was shown that TRIL can interact with TLR4 and that this interaction is enhanced by lipopolysaccharide (LPS) mediated stimulation. This suggests that the regulation of the immune response in IBD mediated by hs-miR-223 could, in turn, involve the regulation of TRIL expression. Controversially, other studies point to a proinflammatory role of hs-miR-223. *Wang* et al. argue through analyzes in animal models that hs-miR-223 interacts with the IL23 pathway targeting CLDN8 (Claudin-8), a critical member of the family of proteins involved in maintaining the normality of the intestinal barrier. According to their theory, hs-miR-223 functions as a proinflammatory miRNA and is tightly controlled by IL23 in IBD [59].

The fact that both hs-miR-155-5p and hs-miR-223-3p are significantly different between cases and controls may be related to their functions described so far, as they are involved in the regulation of hematopoietic cell maturation, being expressed in hematopoietic stem cells, B cells, T cells, monocytes, and granulocytes [60]. The expression of these miRNAs is regulated by the activation of receptors such as TLRs, so potential phenomena of intestinal dysbiosis could trigger activation of the immune response, generating an overexpression of these miRNAs in a transkingdom communication context.

As reported previously, IBD has been associated with a decline in the diversity of commensal microorganisms [61]. However, with respect to UC patients, there is no bibliographical concordance on this point. There is evidence of significant differences in alpha diversity in some populations and an absence of significant differences between UC patients and their healthy partners who share the same environment [62,63]. In our study, no significant differences in alpha diversity between non-UC controls and UC patients were found. Considering that our UC patients were all under medical treatment, the symptoms of inflammation and bleeding were controlled, with 50% of our UC patients under remission and the rest presenting mild clinical features. In this sense, partial or total remission could define a new microbial state, not identical to the pre-pathogenic one but stable and flexible enough to maintain equilibrium in the new inflammatory context. Additionally, beta diversity changed significantly between groups. This would indicate the key role of microbial structure in this disease [62,63]. Regarding differentially abundant taxa, we identified *Bifidobacterium* as the genus found differentially expressed within UC patients compared to non-IBD controls. This is consistent with previous reports that identified *Bifidobacterium* as an overrepresented genus in active UC patients [64]. Some studies suggested that exogenous administration of *Bifidobacterium* strains improved human intestinal barrier function, mucus production, and immune modulation [65,66], thus preventing the circulation of pathogen-associated molecules such as LPS and preventing the development of colitis in mice [67]. The fact that the *Bifidobacterium* genus is overrepresented could explain the controlled inflammatory state that was demonstrated in our UC population, given that half were found to be in clinical remission and the rest with mild activity. Regarding the prediction of differential expression of genes associated with metabolic pathways, menaquinol-9 biosynthesis (associated with vitamin K2 production) was underrepresented in UC patients. Kuwabara et al. had evidenced that IBD patients had a high prevalence of decreased vitamin K and D, probably caused by malabsorption of these vitamins [68]. The fact that this metabolic pathway is underrepresented in UC patients could show that this deficit is not only due to a malabsorption phenomenon but also to a direct effect of the dysbiosis phenomenon associated with UC.

As regards CD patients, both Shannon diversity (alpha diversity) and weighted and unweighted UniFrac distance values (beta diversity) have been statistically different between non-IBD controls and CD patients. Regarding differentially abundant taxa, *Bifidobacterium*, *Bacteroides*, *Lactobacillus*, *Streptococcus*, *Lachnoclostridium*, *Olsenella*, *Faecalibacterium*, *Clostridium_sensu_stricto_1*, *Flavonifractor*, *Turicibacter* are found to be differential only in CD patients compared to non-IBD controls. There are discordant bibliographic reports regarding each of the genders found as differentials in our population, some being equally increased in some populations and diminished in others [69,70,71,72]. It is essential to consider the importance of a global state, where genera should not be evaluated in isolation but rather in the context of a microbial complex signature that can undergo metabolic compensation according to the population structure defined by the pathological state. In this sense, the observed intestinal dysbiosis translated into a trend of increased aromatic degradation, which appears to be overrepresented in many of the metabolic pathways predicted as differential in this group compared to controls without IBD. Gut metabolism is closely related to host health, and the role of aromatic amino acid (AAA) catabolism by the intestinal microbiome is being increasingly evaluated, as it produces numerous metabolites that can regulate immune, metabolic, and neuronal responses in local and distant sites [73]. AAAs are generated by proteolysis in the gastrointestinal tract and are actively sensed and processed by both the host and microbiota. Our results are in line with the bibliography that points to unbiased metabolomic studies that suggested elevated fecal AAAs metabolite levels to discriminate patients with CD from controls [74,75,76]. Targeting AAA metabolism has shown therapeutic promise in animal models of IBD [73].

Additionally, common core microbiota was analyzed, finding that UC and CD have different core microorganisms with different metabolic capabilities compared to the control group and that active or remission states have some interesting differences in terms of core microbial profiles [77,78]. For non-IBD controls, 79 bacterial genera were identified as core microbiota, 65 for UC and 46 genera for CD, respectively. At the intersection of groups, 38 genera were found at the triple intersection, observing only a few core features exclusively represented in each of the groups. With these results, we set out to determine which was the most informative data source to differentiate between cases and controls, whether the information provided by the microbiota or the data from the biochemistry of the patients. Our results showed that both the model built based on biochemical parameters, epigenetic biomarkers, and clinical characteristics, as well as the model built with microbiota data, were sufficient to discriminate controls from patients with IBD. It should be noted that regarding the MM, the genera present in the models for both pathologies do not correspond, with the exception of *Lachnoclostridium*, to those found by the differential abundance test. The fact that differential abundance analysis does not detect such genera is because ANCOM-BC is not ideal for detecting genera that are completely absent from a group. Therefore, by complementing both analyses, differential abundance and core microorganisms, the differences between groups can be characterized in a more comprehensive way in logistic regression models. Furthermore, *Ruminococcus_torques_group*, *Lachnoclostridium*, and *Subdoligranulum* genera were found to be part of the core microbiota of all three groups. Likewise, *Hungatella* formed part of the CD microbiota core, and *Desulfovibrio* was part of the non-IBD control and UC microbiota cores. In this regard, the core microbiome is important for understanding the stable components in complex microbial assemblages. These shared taxa are assumed to represent the most ecologically and functionally important associated microbes in a specific environment. Parameters defined by traditional ecological theory, such as composition, phylogeny, persistence, and connectivity, will also create a portrait of the core microbiome and advance understanding of the role of key microorganisms and their functions within and between ecosystems.

Remarkably, some studies of *Desulfovibrio* strains have shown that intestinal sulfate-reducing bacteria isolated from patients with UC were associated with the development of gut inflammation [79]. Additionally, recent works have explored IgA-SEQ’s ability to identify bacteria with distinct immunomodulatory properties capable of inducing intestinal pathology [80]. By evaluating the fecal microbiome of patients with IBD, Jason M.Shapiro et al. demonstrated a selective enrichment of IgA-coated bacteria, detecting *Lachnoclostridium* among the colitogenic genera for CD [81,82]. In this sense, Zhibing Qiu et al. observed that *Lachnoclostridium* notably increases only in samples from UC patients, and *Subdoligranulum* also decreases significantly, but these changes were not observed in CD samples [83]. Taken together, these data reinforce the diagnostic power of these genera present in our MM for UC.

In line with CD-MM genera, Casey M A Jones et al. have recently associated *Hungatella* with disease activity in patients with CD [84]. Also, the authors point to *Ruminococcus gnavus* as the second most important taxonomic feature in their Random Forest model, considering that although it is among the most common 57 species of the gut microbiome in >90% of people [85], its increased abundance was reported in several cohorts of CD patients [81]. Notably, the PWY5415 pathway (also present in CD-MM) involves catechol degradation, being catechol itself an intermediate in the degradation of many different aromatic compounds. This agrees with the trend of increased aromatic degradation, which appears to be significantly overrepresented in many of the metabolic pathways predicted as differential in our CD group (ANCOM-BC *p* < 0.05).

As regards biochemical models (BM), both hs-miR155-5p and hs-miR223-3p were potent representatives of the pathological state, as in both UC-BM and CD-BM, this miRNA overexpression discriminates accurately between cases and controls. This indicates that, after future validation, the use of these models could simplify daily clinical screening and surveillance in IBD using easy-to-obtain samples such as peripheral blood collection and stool. Notably, it was recently reported that platelets could reflect the severity of Crohn’s disease without the effect of anemia [86]. Even though the primary role of platelets in hemostasis, it has been demonstrated that they have immunological properties as their activation transforms them into high-affinity inflammatory platforms [86,87]. They also express Toll-like receptors (TLRs) that can bind to LPS on the outer membrane of gram(-) bacteria [88] and excrete large amounts of pro-inflammatory substances to activate dendritic cells in the injured tissue [89,90]. In this sense, the authors demonstrated the higher the platelet count, the more serious the CD. Regarding GOT, Yu-Feng Liu et al. have shown how hepatic fibrosis can be aggravated by ulcerative colitis through the activation of hepatic stellate cells (HSCs) and TLR4 signaling through the gut-liver axis [91]. Authors demonstrated in mice models that gut barrier dysfunction in UC leads to bacterial translocation and elevated LPS, which may promote the activation of TLR4 signaling and HSCs in the liver.

Finally, in the present work, we performed a correlation network analysis to gain insight into the interactions between the variables that are part of the selected logistic regression models for UC and CD patients. In a detailed view of Figure 9 and Figure 10, it can be noticed that Hungatella and miR155_MF from the CD network, as well as Desulfovibrio, platelets, and miR223_MF from the UC network, are variables from the respective logistic regression models that are not shown. This is because these variables were filtered out from the networks due to their relatively weak interaction with other nodes (r < |0.7|). In this matter, it should be clarified that, with the exception of Hungatella, all these variables were part of the network built for the non-IBD control group (data not shown) even after applying the correlation filters. In fact, the weighted correlation networks of the control group presented close interactions, where all the edges represented coefficients greater than |0.7| and nodes associated in 7 clusters only. The disintegration of interactions observed in the pathological situations with respect to the control group could suggest that, in the context of IBD, correlations are probably lost or shifted because other biological processes underlying the disease gain influence on certain variables. This reinforces the concept of the complexity that characterizes these multifactorial diseases and the need for biomarkers capable of fully outlining the main processes that underlie the pathogenesis of each patient.

Notably, this is the first report describing the epigenetic and microbial taxonomic composition of intestinal microbiota in Argentine subjects diagnosed with IBD. Although the vast majority of studies of intestinal microbiome composition in different human populations are performed from fecal samples, it is important to note that human fecal microbiota is not a faithful reflection of the cecal or colonic microbiota [92]. Despite this, the use of this sampling methodology is less invasive than endoscopies and biopsies that can lead not only to health but also to ethical problems since not only sick patients but also healthy volunteers are intervened. Therefore, the study of fecal matter is a limitation that must be considered in the interpretation of the results, but not an exclusion. Additionally, it is important to note that this study is correlational, and therefore, conclusions related to the pathogenesis of IBD at these low taxonomic levels are difficult. Even though the sample size could be improved, this pilot study contributed to the knowledge of the uncharacterized miRNA biomarkers and gut dysbiosis associated with IBD patients in the Argentinean population. Similar studies have been conducted in small cohorts and the changes observed were clear [71]. Further functional characterization, such as proteomics or metabolomics, as well as longitudinal metagenomic shotgun studies, should be performed in South American IBD patients. This could improve sampling limitations and consider intestinal mucosa metabolism within our local environmental factors, allowing a better understanding of the role of gut dysbiosis in these chronic diseases of unknown etiology [13,14,15,93].

Overall, our analysis of specific epigenetic and microbial signatures related to the transkingdom development of IBD in our region provides a first exploration of the IBD-associated features in Buenos Aires (BA) and its metropolitan area, which constitutes a megalopolis being the second most populated agglomeration in South America and the southern hemisphere.

## 5. Conclusions

In this work, we set out to integrate data of uncharacterized epigenetic biomarkers and gut microbiota profiles of IBD patients in our region, with relevant clinical and biochemical features in the follow-up of patients, with the aim of generating more powerful diagnostic tools to discriminate phenotypes. Overall, our study provides new insights into data analysis algorithms to promote comprehensive phenotyping tools using quantitative and qualitative analysis in a transkingdom interactions network context.

## Figures and Tables

**Figure 1 microorganisms-10-02190-f001:**
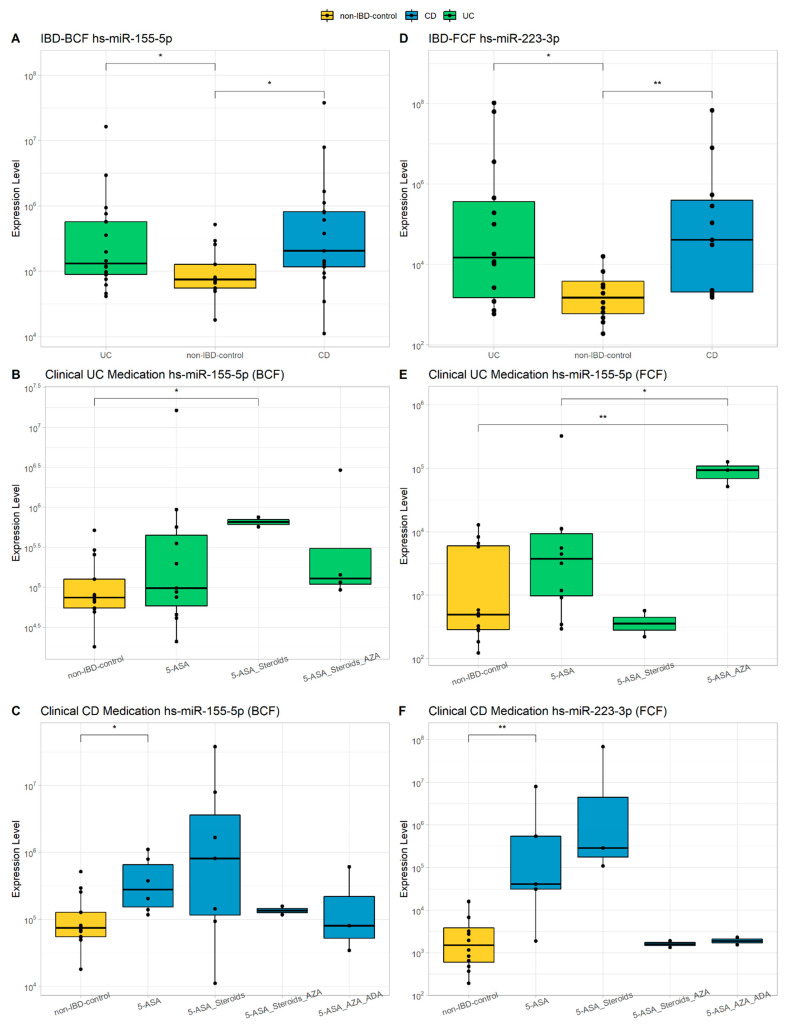
Evaluation of miRNAs. hs-miR-155-5p and hs-miR-223-3p expression levels in IBD patients compared to non-IBD controls (**A**,**D**). Expression of hs-miR-155-5p and hs-miR-223-3p compared to non-IBD controls in relation to medication received by UC patients or CD patients (**B**,**C**,**E**,**F**). Wilcoxon test was calculated between groups (* = *p*-value < 0.05, ** = *p*-value < 0.01). 5-ASA, 5-aminosalicylic acid (mesalamine); AZA, azathioprine; ADA, Adalimumab biological therapy.

**Figure 2 microorganisms-10-02190-f002:**
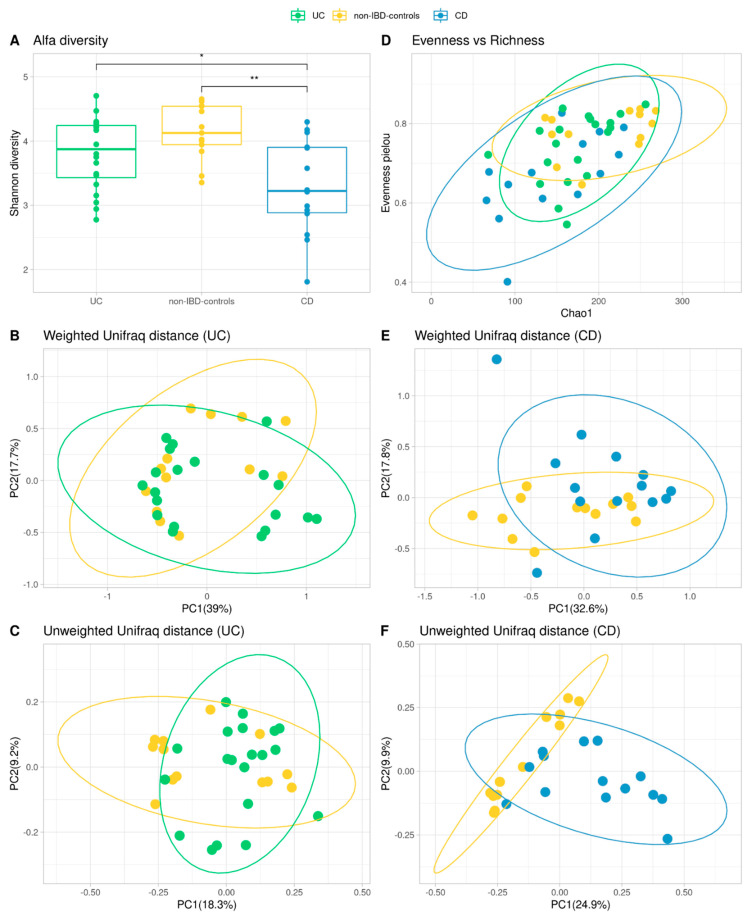
Comparison of microbiota community of the IBD patients and non-IBD control groups. Shannon diversity measures for Alpha Diversity (**A**,**D**) and Wilcoxon test were calculated between groups (* = *p*-value < 0.05, ** = *p*-value < 0.01) (**A**). Representation of richness and evenness in a dot plot (**D**). PCoA plots of beta diversity (Weighted and Unweighted Unifrac distances) for UC patients and non-IBD controls (**B**,**C**) and CD patients and non-IBD controls (**E**,**F**).

**Figure 3 microorganisms-10-02190-f003:**
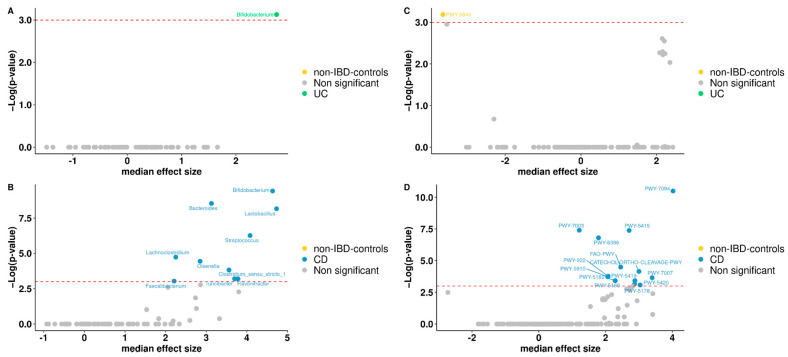
Volcano plot. Differentially abundant genera between non-IBD controls and UC patients (**A**) or non-IBD controls and CD patients (**B**). Volcano plot of the differentially abundant metabolisms between non-IBD controls and UC patients (**C**) or non-IBD controls and CD patients (**D**). Red dash lines represent a significance threshold (*p*-value = 0.05). Yellow, green or blue represents significant genera or metabolisms more abundant. In gray, non-significant features are represented.

**Figure 4 microorganisms-10-02190-f004:**
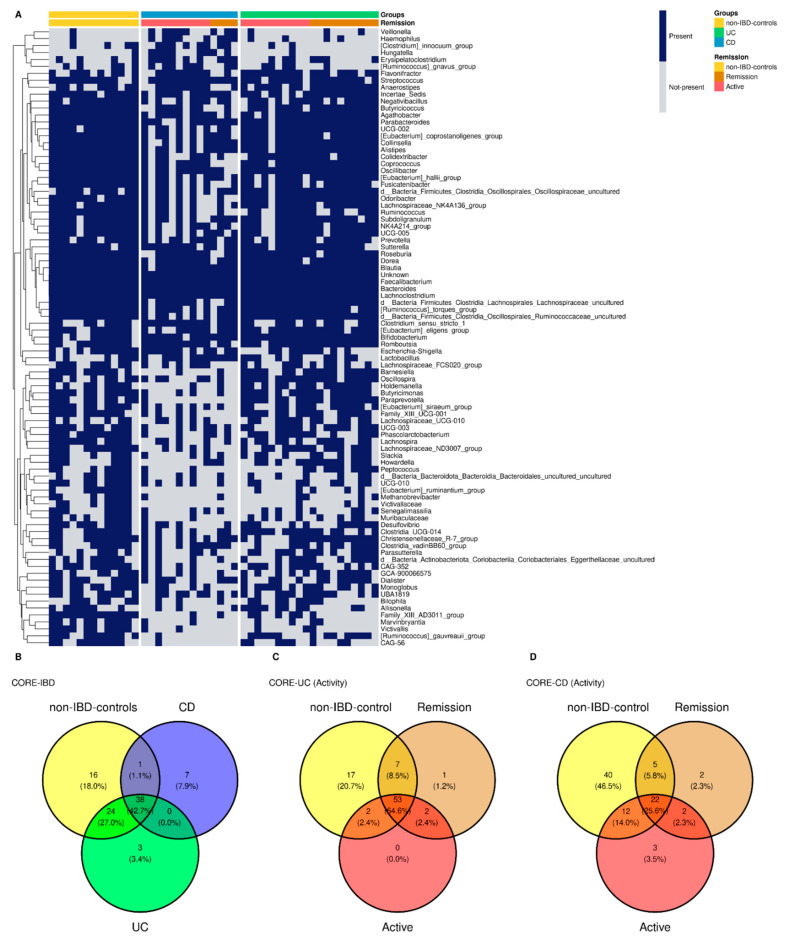
Heatmap core. Genera core microbiota for non-IBD controls and IBD groups. The heatmap also distinguishes active/remission within the IBD group (**A**). Venn diagram represents shared/exclusive core genera between groups (**B**). Venn core microbiota for UC patients (**C**) and CD patients (**D**) discerning active and remission patients.

**Figure 5 microorganisms-10-02190-f005:**
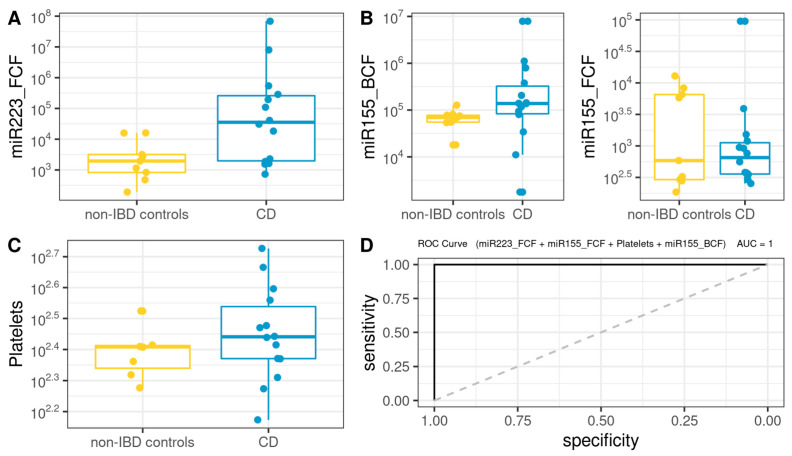
Box plot in the biochemical model. Values of the variables present in patients with CD compared to non-IBD controls (**A**–**C**). Performance of CD biochemical model, as assessed via the area under the ROC curve (**D**).

**Figure 6 microorganisms-10-02190-f006:**
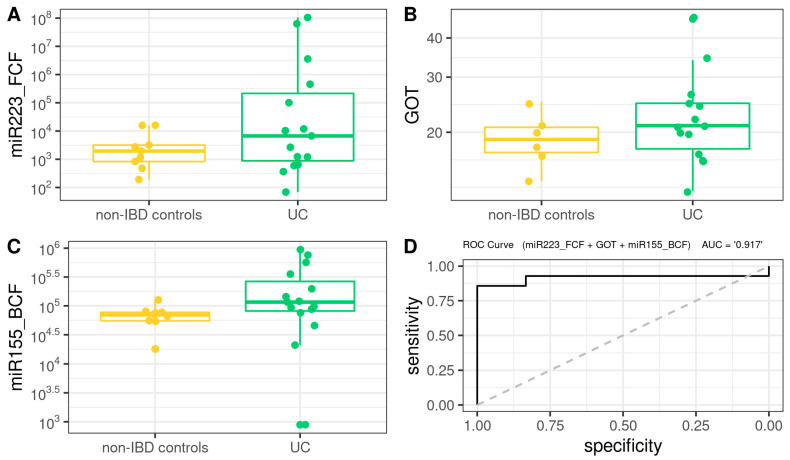
Box plot in the biochemical model. Values of the variables present in patients with UC compared to non-IBD controls (**A**–**C**). Performance of biochemical model, as assessed via the area under the ROC curve (**D**).

**Figure 7 microorganisms-10-02190-f007:**
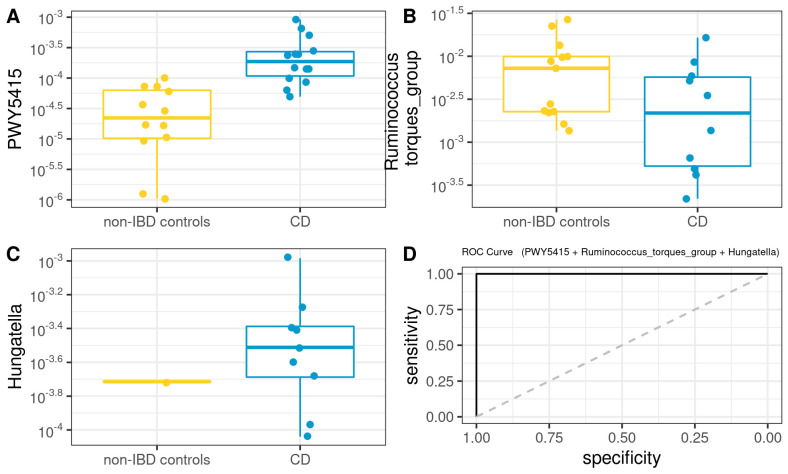
Box plot in the microbiota model. Values of the variables present in patients with CD compared to non-IBD controls (**A**–**C**). Performance of microbiota model, as assessed via the area under the ROC curve (**D**).

**Figure 8 microorganisms-10-02190-f008:**
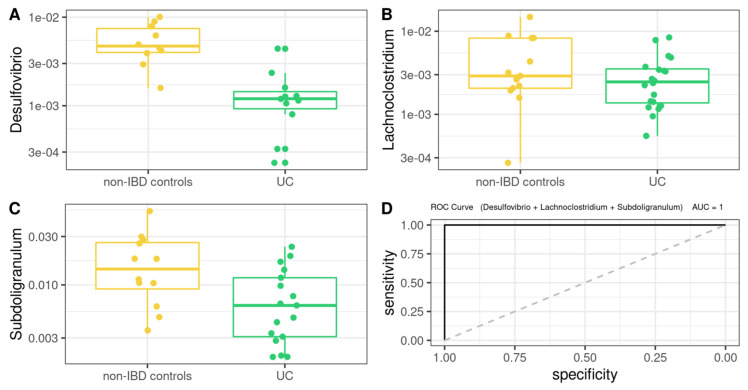
Box plot in the microbiota model. Values present in patients with UC compared to non-IBD controls (**A**–**C**). Performance of microbiota model, as assessed via the area under the ROC curve (**D**).

**Figure 9 microorganisms-10-02190-f009:**
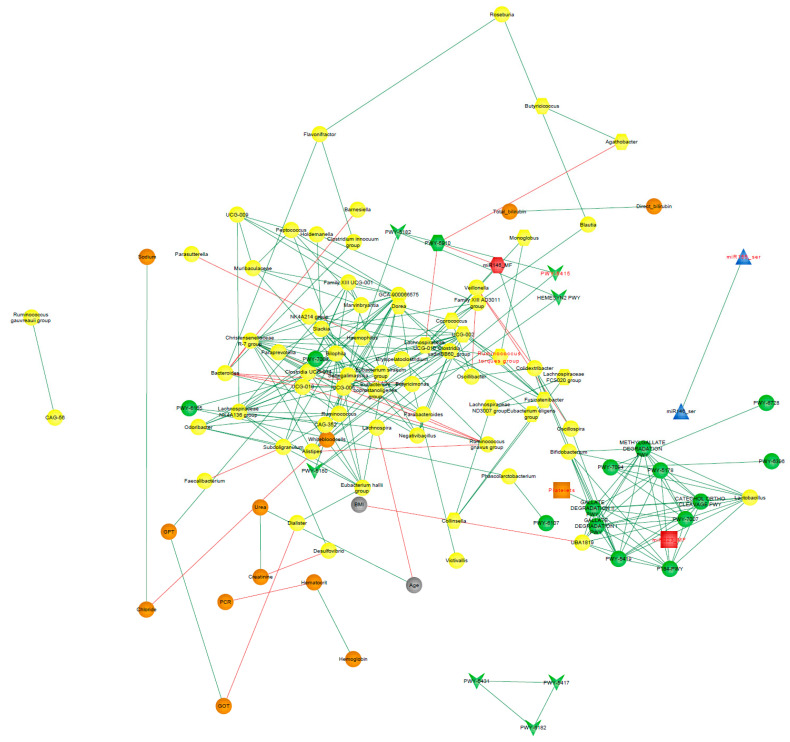
Correlation CD-network. Clinical characteristics age, BMI (gray nodes), biochemical measurements (orange nodes), core microbiota and differentially abundant bacterial taxa (yellow nodes), statistically significant differentially abundant metabolic pathways (*p* < 0.05; green nodes), and miRNAs from BCF (blue nodes) and FCF (red nodes) of the CD patients. Only correlations with Spearman’s correlation coefficients over 0.7 are shown. Green edges represent positive correlations. Red edges represent negative correlations. Variables from the logistic regression model of CD patients are indicated in red font.

**Figure 10 microorganisms-10-02190-f010:**
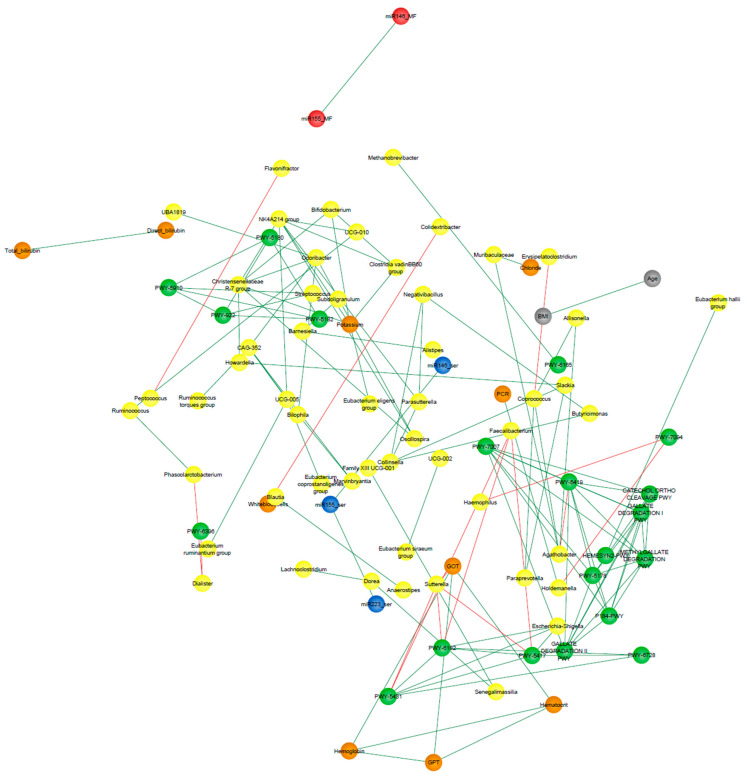
Correlation UC-network. Clinical characteristics, age, BMI (gray nodes), biochemical measurements (orange nodes), core microbiota and differentially abundant bacterial taxa (yellow nodes), statistically significant differentially abundant metabolic pathways (*p* < 0.05; green nodes), and miRNAs from BFC (blue nodes) and FCF (red nodes) of the UC patients. Only correlations with Spearman’s correlation coefficients over 0.7 are shown. Green edges represent positive correlations. Red edges represent negative correlations. Variables from the logistic regression model of UC patients are indicated in red font.

**Table 1 microorganisms-10-02190-t001:** Population general description.

Groups		UC	CD	Non-IBD Controls	*p*
General descriptions	Overall subjects	n = 20	n = 14	n = 13	-
	Female %	60	57	69	ns
	Male %	40	43	31	ns
	Mean age, years ± (SD)	46.65 ± (17.80)	44.42 ± (16.61)	52.84 ± (17.94)	ns
	BMI ± (SD)	26.60 ± (4.09)	25.01 ± (3.84)	28.40 ± (5.44)	ns

Variables were assessed by ANOVA test. No significant (ns).

**Table 2 microorganisms-10-02190-t002:** IBD patients’ characterization.

Groups		UC	CD
		mean ± (SD)	
Biochemical data	Hemoglobin, g/dL	13.74 ± (1.41)	12.61 ± (1.92)
	us-CRP, mg/dL	0.35 ± (0.26)	1.37 ± (2.48)
	Albumin, g/dL	4.55 ± (0.34)	4.39± (0.25)
	Platelets, ×10^3^/^mL^	238.11 ± (58.02)	297.86 ± (107.15)
	FPG, mg/dL	92.43 ± (13.36)	97.35 ± (15.98)
	Creatinine, mg/dL	0.77 ± (0.16)	0.82 ± (0.22)
	Triglycerides, mg/dL	92.47 ± (49.33)	107.22 ± (43.28)
	Total cholesterol, mg/dL	177.77 ± (43.04)	216.16 ± (53.09)
	LDL-C, mg/dL	109.07 ± (42.50)	136.89 ± (41.31)
	HDL-C, mg/dL	56.87 ± (13.44)	61.50 ± (22.45)
	GOT (UI/I)	23.89 ± (8.49)	23.89 ± (8.49)
	GPT (UI/I)	22.14 ± (10.88)	23.14 ± (10.88)
		No. (%)	
IBD therapy	5-ASA	20 (100.00)	14 (100.00)
	Steroids	1 (5.00)	5 (35.71)
	AZA	4 (20.00)	3 (21.42)
	Rectal budesonide	5 (25.00)	4 (28.57)
	ADA	0 (0.00)	2 (14.28)
Lesion localization UC	E1 proctitis	1 (5.00)	-
	E2 left-sided colitis	8 (40.00)	-
	E3 extensive	11 (55.00)	-
Lesion localization CD	L1 ileal	-	0 (0.00)
	L2 colon	-	8 (57.14)
	L3 ileocolonic	-	4 (28.57)
	L2-L4 upper GIT	-	1 (7.14)
	L3-L4 upper GIT	-	1 (7.14)
General activity according to medical criteria	General remission	10 (50.00)	4 (28.57)
	General active	10 (50.00)	10 (71.42)
Clinical activity (Truelove & Witts and CDAI)	Remission	15 (75.00)	10 (71.42)
	Mild	5 (25.00)	4 (28.57)
	Moderate	0 (0.00)	0 (0.00)
	Severe	0 (0.00)	0 (0.00)
Endoscopic score (Mayo score and SES-CD)	Normal	10 (50.00)	4 (28.57)
	Mild	7 (35.00)	4 (28.57)
	Moderate	1 (5.00)	3 (21.42)
	Severe	2 (10.00)	3 (21.42)
Histology activity	Quiescent	10 (50.00)	3 (21.42)
	Inflammatory infiltrate	10 (50.00)	11 (78.57)

Ultra sensitive C-Reactive Protein: us-CRP, Fasting plasma glucose: FPG, 5-ASA: Mesalazine, AZA: Azathioprine, ADA: Adalidumab, glutamic oxaloacetic transaminase: GOT, glutamic pyruvic transaminase: GPT.

## Data Availability

Raw sequences of the 16S rRNA gene reported in this article were deposited in NCBI Short Read Archive (SRA) and are accessible under the accession number PRJNA646271.

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
