# Peer review of "Comprehensive Phenotyping in Inflammatory Bowel Disease: Search for Biomarker Algorithms in the Transkingdom Interactions Context"

_microorganisms, 2022, doi:10.3390/microorganisms10112190_

Round 1
Reviewer 1 Report
The manuscript by Ayelen Daiana presents extremely valuable data regarding IBD patients. However, a few points require clarification.
1. The abbreviations TGO and FMI need to be defined.
2. Figure 5D shows an ROC curve for miR223_FCF + miR155_FCF + Platelets + miR155_BCF. The data for miR155_FCF used to generate this curve needs to be displayed.
Author Response
- The abbreviations TGO and FMI need to be defined.
* FMI was replaced by IMF which corresponds to the abbreviation for intestinal myofibroblasts. TGO was replaced by GOT which corresponds to the abbreviation for glutamic pyruvic transaminase. Both abbreviations were highlighted in red and referenced in the text with their corresponding meaning. In addition Figures 6 and 10 were also modified in order to replace TGO with GOT.
- Figure 5D shows an ROC curve for miR223_FCF + miR155_FCF + Platelets + miR155_BCF. The data for miR155_FCF used to generate this curve needs to be displayed.
* Data corresponding to miR155_FCF was added to Figure 5B.
Reviewer 2 Report
The manuscript entitled "Comprehensive phenotyping in inflammatory bowel disease: search for biomarker algorithms in the transkingdom interactions context", is original and current. To improve the study, I recommend avoiding the frequent use of the word "we" and improving Figures 9 and 10 by avoiding overlapping writing.
Author Response
The manuscript entitled "Comprehensive phenotyping in inflammatory bowel disease: search for biomarker algorithms in the transkingdom interactions context", is original and current. To improve the study, I recommend avoiding the frequent use of the word "we" and improving Figures 9 and 10 by avoiding overlapping writing.
* The frequent use of “we” was reconsidered and the respective correction in each case was highlighted in red in the text. Additionally, Figures 9 and 10 were improved by avoiding overlapping writing.
Reviewer 3 Report
- Use oxford comma
- Use bold only for column titles in tables and not for column content
- “Irritable”
Use lower case
- “Rutgeerts score [22] for patients who have undergone surgery”
You have to use Rutgeerts score only for colonoscopy 1 year after surgery
- Do not use bold for “Logistic regression model” content
- How did you calculate the sample size for your study?
- In addition to the crude number, add the percentage in Table 2
- Some of the data are reported twice in Table 2
- Why did you not include patients with moderate or severe clinical activity?
- “The use of miRNAs as biomarkers is a widely studied strategy worldwide in various
complex pathologies with an inflammatory component due to the availability of massive
public data on gene expression.”
In which clinical settings miRNAs are used?
- Use the same character in “Common core microbiota” paragraph
- “but also highlights”
do not go to line
Author Response
Comments and Suggestions for Authors:
- Use oxford comma: We have contemplated this suggestion throughout the text. Oxford commas were highlighted in red.
- Use bold only for column titles in tables and not for column content: * done ✔️
- “Irritable” Use lower case: * done ✔️
- “Rutgeerts score [22] for patients who have undergone surgery”. You have to use Rutgeerts score only for colonoscopy 1 year after surgery:
Considering the reviewer's suggestion, we searched the clinical history of each case for the exact date of surgery of the patients that we had classified with the Rutgeerts score. Indeed, the 4 patients had exceeded 12 months post-surgery, so it was decided to reclassify them with the SES-CD endoscopic scale according to medical criteria. This is reflected in Table 2 of the revised manuscript.
- Do not use bold for “Logistic regression model” content: * done ✔️
- How did you calculate the sample size for your study? La Rosa et al. showed that the Dirichlet-multinomial distribution can be used to calculate power and sample sizes for experimental design and to perform tests of hypotheses for microbiota assays (2012 - Hypothesis testing and power calculations for taxonomic-based human microbiome data. PLoSOne, 7, e52078). With an expected number of more than 50,000 sequence reads per sample and an α of 5%, ~15 subjects per group were required for ~80% power. To specify this, the following paragraph was added to the ‘Selection of participants and environmental data’: The sample size was determined using a Dirichlet multinomial distribution model. With an expected number of more than 50,000 sequence reads per sample and an α of 5%, 15 subjects per group were required for ~80% power [17].
- In addition to the crude number, add the percentage in Table 2: * done ✔️
- Some of the data are reported twice in Table 2: * done ✔️
- Why did you not include patients with moderate or severe clinical activity?
In the present work we have included patients attending the Gastroenterology service of Hospital Posadas only. This service is frequently attended by chronic patients, who are usually in clinical remission or at least with controlled activity. Therefore, unfortunately, we were unable to include and analyze patients with moderate or severe clinical activity during the participant recruitment period. The patients analyzed are those who attended a medical consultation at the time planned for the sampling.
- “The use of miRNAs as biomarkers is a widely studied strategy worldwide in various complex pathologies with an inflammatory component due to the availability of massive public data on gene expression.” In which clinical settings miRNAs are used?
We have added some of the bibliographic citations that support the statement.
- Use the same character in “Common core microbiota” paragraph: * done ✔️
- “but also highlights” do not go to line: * done ✔️
Finally, we have changed the supplementary data file, which had been uploaded in word format. The current file is an excel, since we consider this format to be more suitable for a better visualization of the data.
Round 2
Reviewer 3 Report
- Always use the same sign for the decimal digits (see 15.0 and 0.22 in Table 2)
- “The use of miRNAs as biomarkers is a widely studied strategy worldwide in various complex pathologies with an inflammatory component due to the availability of massive public data on gene expression.” In which clinical settings miRNAs are used?
We have added some of the bibliographic citations that support the statement.
These references do not prove that miRNAs are used in clinical practice
-
Author Response
Reviewers comments and the respective corrections are detailed below:
- Always use the same sign for the decimal digits (see 15.0 and 0.22 in Table 2) * done ✔️
- “The use of miRNAs as biomarkers is a widely studied strategy worldwide in various complex pathologies with an inflammatory component due to the availability of massive public data on gene expression.” In which clinical settings miRNAs are used? We have added some of the bibliographic citations that support the statement. These references do not prove that miRNAs are used in clinical practice.
* We modified these references. The new citations correspond to [44-48]:
- Circulating microRNAs: Biomarkers of disease. Clin Chim Acta. 2021;516: 46–54.
- Galvão-Lima LJ, Morais AHF, Valentim RAM, Barreto EJSS. miRNAs as biomarkers for early cancer detection and their application in the development of new diagnostic tools. Biomed Eng Online. 2021;20: 21.
- Okuda Y, Shimura T, Iwasaki H, Fukusada S, Nishigaki R, Kitagawa M, et al. Urinary microRNA biomarkers for detecting the presence of esophageal cancer. Sci Rep. 2021;11: 8508.
- Masi L, Capobianco I, Magrì C, Marafini I, Petito V, Scaldaferri F. MicroRNAs as Innovative Biomarkers for Inflammatory Bowel Disease and Prediction of Colorectal Cancer. Int J Mol Sci. 2022;23. doi:10.3390/ijms23147991
- Mi S, Zhang J, Zhang W, Huang RS. Circulating microRNAs as biomarkers for inflammatory diseases. Microrna. 2013;2: 63–71.
We thank the editors and reviewers for their thoughtful comments that helped us improve the quality of our manuscript.
